# The Mode of Activity of Cervical Extensors and Flexors in Healthy Adults: A Cross-Sectional Study

**DOI:** 10.3390/medicina58060728

**Published:** 2022-05-28

**Authors:** Hiroyoshi Yajima, Ruka Nobe, Miho Takayama, Nobuari Takakura

**Affiliations:** Department of Acupuncture and Moxibustion, Tokyo Ariake University of Medical and Health Sciences, Tokyo 135-0063, Japan; yajima@tau.ac.jp (H.Y.); r.nobe@tau.ac.jp (R.N.); takayama@tau.ac.jp (M.T.)

**Keywords:** non-specific neck pain, cervical muscles, surface electromyogram, flexion relaxation phenomenon, craniovertebral angle, cervical vertebrae, neck muscles, electromyography

## Abstract

*Background and Objectives:* The purpose of this study was to investigate the activity of bilateral cervical extensors and flexors on the sagittal, frontal, and horizontal planes of healthy adults during motions of the neck in a sitting position, which has not been satisfactorily investigated by surface electromyogram (sEMG). *Material*
*and Methods:* We recruited 35 healthy participants (mean ± standard deviation of age, 20.3 ± 2.4). sEMG recordings of the cervical extensors and flexors were performed for a total of nine seconds in three phases: Phase I involved the motion of the neck from the neutral position to the maximum range of motion; Phase II involved maintaining the neck at the maximum range of motion; and Phase III involved the motion of the neck from the maximum range of motion to the neutral position during neck flexion, extension, right and left lateral flexion, right and left rotation, and maintaining the neck in the neutral position. Muscle activities in each motion were normalized as a percentage of maximal voluntary contraction (%MVC) so that the muscles could be compared. *Results:* The %MVC of the extensors was significantly larger than that of the flexors in the neutral position (*p* < 0.001). In addition, the %MVCs of the following were significantly larger than the %MVC in the neutral position: the extensors in flexion (*p* = 0.014) and extension (*p* = 0.020), the ipsilateral extensors (*p* = 0.006) and flexors (*p* < 0.001) in lateral flexion in Phase I; the flexors in flexion (*p* < 0.001), the extensors in extension (*p* = 0.010), and the ipsilateral extensors and flexors in lateral flexion (*p* < 0.001) in Phase II; the extensors and flexors in flexion (*p* < 0.001), the flexors in extension (*p* < 0.001), the ipsilateral flexors (*p* < 0.001), the contralateral flexors (*p* = 0.004) and the contralateral extensors (*p* = 0.018) in lateral flexion in Phase III; and the bilateral extensors and contralateral flexors during rotation in all three phases (*p* < 0.001). *Conclusion:* The typical sEMG activities of the extensors and flexors during motion of the neck in healthy adults were identified in this study; this information can be used to understand the pathophysiology of non-specific neck pain and to provide an index for evaluating the effect of treatment.

## 1. Introduction

Non-specific neck pain (NSNP) is defined as pain in the posterior and lateral aspect of the neck, between the superior nuchal line and the spinous process of the first thoracic vertebra [1], lasting for more than three months without any specific causes [2]. Two-thirds of people experience NSNP at some stage, especially in middle age [3], which has been a global burden with a significant impact on our society [4,5,6,7]. The pathoanatomical reason for NSNP is not well understood, although mechanical factors, such as thickening of the trapezius fascia [8], microcirculatory disturbances of the trapezius muscle [9], altered activity of the cervical muscles [10], and poor head posture [11] are likely involved in causing chronic neck pain [3,12].

The cervical structure is required for both mechanical flexibility, which gives rise to a broad range of motion, and stability in bearing the weight of the head. These two characteristics are supported by constant well-coordinated activities of the cervical muscles [13], which provide nearly 80% of the mechanical stability of the cervical spine [14]. Flexibility of the neck facilitates a wide range of motion in the three anatomical planes, the sagittal, frontal, and horizontal [15]. Considering such characteristics of the cervical structure, a delicate imbalance of the activities of the cervical muscles, especially the activities of the extensor and flexor muscles, impairs the sound flexibility and stability of the neck, resulting in various complaints about the neck region, as observed in NSNP patients [16]. A decrease in the strength and endurance capacity of the cervical extensor and flexor muscles in patients with neck pain was reported in many studies [17,18,19,20,21].

Surface electromyography (sEMG) is a useful tool in revealing the mechanisms or causes of NSNP. Significantly larger activities of the cervical muscles [5,22] and deficits in the motor control of cervical flexor muscles [23] were reported in the sEMGs of people with neck pain with isometric contraction, compared to no-pain control participants. In studies using isotonic contraction in NSNP patients, the flexion-relaxation phenomenon, which is the electrical silence of cervical extensors in sEMG observed during full cervical flexion in healthy participants [24], was not observed [25,26]. There was also an increase in the sEMG activities of the superficial cervical flexor muscles in NSNP patients, compared with the asymptomatic control patients, during a craniocervical flexion test [27,28]. These studies showed that sEMG is a useful index in investigating the mechanisms of neck pain. However, until now, there have been no basic sEMG data associated with isomeric or isotonic contraction in the bilateral extensors and flexors with flexion, extension, lateral flexion, and rotation of the neck in patients with neck pain. Such data may lead to an understanding of the pathophysiology of neck pain and to an index for evaluating the effectiveness of treatments.

Although isometric contraction has been used as an index of neck muscle activity in many studies [4,5,19,22,29,30,31], investigations that observed the isotonic contraction of the cervical muscles [24,25,26], which occurs mainly in the daily activities of healthy people, were scarce. In this study, using sEMG, we measured the muscle activity in the neck at the neutral position and during isotonic contraction (during motions) in healthy participants to obtain the basic data of cervical muscle activity to provide a basis for an objective evaluation of NSNP. To the best of our knowledge, no study has previously examined the sEMG activity of the cervical extensors and flexors due to motions in the sagittal, frontal, and horizontal planes of the neck in people without neck pain. In other words, at present, even if the sEMG activity of the cervical extensors and flexors in patients with neck pain is determined in clinical practice, it might not provide any information to reveal the cause of pain because of the lack of basic data from healthy people. For this reason, we observed the sEMG activity of the cervical muscles in people without neck pain, which provided information that can be used in future for an essential comparison in understanding the pathophysiology of NSNP and in determining the effect of treatments. By dividing the neck muscles into the extensors and flexors, the introduction of sEMG for the evaluation of neck pain in clinical practice can be simplified, and this method will lead to an objective evaluation of neck motions in daily life. The novelty of this study lies in the fact that it was easier to introduce sEMG into clinical practice and to obtain more objective evaluation reflecting daily activity than it was to use the indices of isometric contraction.

The recording of sEMG activities with isotonic contraction is easier to introduce into the clinical field for the evaluation of neck pain than isometric sEMG, which requires resistance with special equipment or by hand [5,22]. In this study, we investigated sEMG with isotonic activities of the cervical extensors and flexors at the neutral position and during neck flexion, extension, lateral flexion, and rotation in healthy participants without any neck complaints to obtain basic information needed to understand the mechanisms of neck pain and to evaluate the effect of treatments on neck pain, e.g., pharmacotherapy, such as muscle relaxants, and physical therapy, such as acupuncture. Those evaluations might be achieved by comparing the results obtained for the muscle activity of the healthy participants in this study with the results obtained for patients with neck pain before and after treatment.

The null hypothesis in this study was that there were no differences in the sEMG activity in the following neck positions: (1) between the extensors and flexors at the neutral position; (2) between the neutral position and neck flexion, extension, lateral flexion, and rotation, respectively, for the extensors and flexors; and (3) between the extensors and flexors during neck flexion, extension, lateral flexion, and rotation.

## 2. Methods

The Ethics Committee of Tokyo Ariake University of Medical and Health Sciences approved this study (ethics approval: Tokyo Ariake University of Medical and Health Sciences no. 227, date of approval: 21 January 2019). The study design was a cross-sectional study. All data were recorded in a shielded room in a laboratory of Tokyo Ariake University of Medical and Health Sciences. Participant recruitment began in January 2019 and all data were obtained from January 2019 to June 2021.

### 2.1. Participants

Thirty-five healthy participants, with a mean ± standard deviation of age, 20.3 ± 2.4, who had no neck and shoulder pain or history of experiencing such pain, were recruited; they signed an informed consent form after the purpose and methods of this study were explained to them. Before each participant was entered in this study, an assistant checked to see if they met the acceptance criteria. The inclusion and exclusion criteria were as follows.

Inclusion criteria: participants with no underlying diseases, no past or present pain in the neck, shoulders, or upper extremities, and no neurological symptoms such as numbness or hypoesthesia.Exclusion criteria: participants who were aware of pain in the neck, shoulders, and upper extremities, or had neurological symptoms such as numbness or hypoesthesia, or who suffered from diseases such as cervical disc herniation or rheumatoid arthritis.

### 2.2. Procedures for Recording Cervical Muscle Activity

#### 2.2.1. Surface Electromyogram (sEMG)

We recorded the sEMG (Neuropack X1: MEB-2306, NIHON KOHDEN CORPORATION, Tokyo, Japan) of the bilateral cervical extensors and flexors. After disinfecting the skin, pairs of disposable Ag/AgCl surface electrodes (NSC electrode, NM-317Y3, NIHON KOHDEN CORPORATION, Tokyo, Japan) positioned 2 cm apart were applied bilaterally to a point approximately 2 cm lateral from the 4th cervical spinous process for the cervical extensors [26,32], and to a point one-third of the line from the mastoid process connecting to the suprasternal notch for the cervical flexors [27,29]. A reference electrode was put on the clavicle and the spine of the scapula. To remove the bias caused by different positions of the electrodes of an electromyogram attached to each participant, one assistant attached all electrodes to all participants and the researcher reconfirmed it to minimize the variation in location of electrodes among participants. sEMG signals were sampled at 1000 Hz, and a bandpass filter between 20 and 500 Hz was used [27,30]. Recorded signals were input into data analysis software (LabChart Pro8, ADInstruments Japan, Nagoya, Japan), and they were fully wave-rectified.

#### 2.2.2. Flow of sEMG Measurement

Participants were seated on a chair with a straight back, bending their hips and knees at 90 degrees and with their feet on the floor. They kept their heads and necks at 0 degrees, eyes straight ahead, pulled their chins, and relaxed their hands on their upper thighs [33,34,35]. We defined this position as the neutral position.

sEMG recordings of the cervical extensors and flexors in participants were performed for nine seconds in the following order: with the neck remaining in the neutral position; flexion in the sagittal plane; extension in the sagittal plane; right lateral flexion in the frontal plane; left lateral flexion in the frontal plane; right rotation in the horizontal plane; and left rotation in the horizontal plane.

After measurements of all motions above, the maximal isometric voluntary contraction (MVC) was measured in the neutral position against manual resistance by a research assistant for five seconds [36] in each of the six motions to normalize sEMG amplitude with the cervical motions obtained in the three anatomical planes.

Each sEMG recording in the neutral position and the six motions consisted of the following three phases of three seconds: a motion from the neutral position to the maximum range of motion (Phase I); maintaining the neck position at the maximum range of motion (Phase II); a motion from the maximum range of motion to the neutral position (Phase III) [25,26,34]. The participants performed these motions on the assistant’s cues every second. sEMGs were recorded three times for the neutral position and for each of the six motions, and the MVC for each direction was measured with two minutes rest between recordings to prevent muscle fatigue.

### 2.3. Assessment of Head and Neck Posture

The head and neck posture in the sagittal plane of the participants sitting in a chair with the neutral position was evaluated by pictures taken with a digital camera (D5300, NIKON CORPORATION, Tokyo, Japan) placed at the level of the participants’ shoulders on the left side of the participants, 1.5 m apart [37,38]. Two markers were placed at the left tragus of the ear and the spinous process of the 7th cervical vertebra (C7) as reference points to measure the craniovertebral angle (CVA) of the participants. Three images were taken for each participant.

### 2.4. Data Analysis

#### 2.4.1. Grand Ensemble Average of sEMGs in Each Motion

For individual participant data, ensemble averages of sEMGs on both sides of the cervical extensors and flexors in the neutral position and in each of the six motions were generated by three rectified sEMGs that were converted from the three sEMGs. For a grand ensemble average of compiling the sEMGs of all participants, the individual ensemble averages of sEMGs were used for the same muscles for the neutral position and for each of the six motions [39,40,41]. The grand ensemble average was used to depict the average activity curve of each of the bilateral extensors and flexors in the neutral position and in each motion.

#### 2.4.2. A Percentage of Maximal Voluntary Contraction (%MVC) of Cervical Muscles

An integrated EMG (iEMG) value was obtained from sEMGs on both sides of the cervical extensors and flexors during each phase in the neutral position and in each of the six motions, and three iEMG values in the same phase in the same muscles were averaged. iEMG values in the bilateral cervical extensors and flexors during the middle three seconds of a five-second sEMG recording of the MVC for flexion, extension, right lateral flexion, left lateral flexion, right rotation, and left rotation were obtained. Three iEMG values in the same muscles in the same motion during the three seconds were averaged [42,43,44]. Then, an iEMG value of each phase of the six motions and the neutral position on each side of the cervical extensors and flexors was divided by an iEMG value of the MVC of the corresponding muscles to obtain a percentage of maximal voluntary contraction (%MVC) for normalization. The %MVCs were regarded as representative values of muscle activities in each phase of each motion.

For extension and flexion, both sides of the cervical extensors and flexors were combined. For lateral flexion and rotation, cervical extensors and flexors activities in both the right and left motions were combined as the ipsilateral and contralateral side.

#### 2.4.3. Craniovertebral Angle (CVA)

We measured the CVA, which is an angle between a horizontal line passing through the C7 spinous process and a line joining the midpoint of the tragus of the ear to the C7 spinous process [11,37,38,45], on a photograph of the participant’s face taken from the left side using ImageJ (NIH), as an objective method to assess head and neck posture [11]. A mean value of the CVAs obtained from three pictures of each patient was used for analysis [45,46].

### 2.5. Statistical Analysis

Statistical analyses were performed using IBM SPSS Statistics version 27 (IBM Japan, Ltd., Tokyo, Japan). A paired T-test was used to compare the %MVC of the extensors and flexors in each phase in the motions. Correlations between the CVA and the %MVC of the second phase in the neutral position were estimated by Pearson’s correlation coefficient. Comparisons of the %MVC between the extensors and flexors for the neutral position and for each phase in the motions were performed using a one-way repeated-measures analysis of variance (ANOVA), and Dunnett tests were performed post hoc when a significant difference was found. ANOVA and Bonferroni tests were used for comparisons of the %MVC among the six motions. The statistical significance was set at *p* < 0.05.

## 3. Results

### 3.1. Activities in the Cervical Extensors and Flexors in the Neutral Position and Craniovertebral Angle (CVA)

The grand ensemble averages of the sEMGs of the cervical extensors and flexors in each motion are shown in Figure 1. The extensors and flexors were activated in the neutral position.

Figure 2 shows the %MVC (mean ± standard deviation) of the cervical extensors and flexors in each phase of each motion, and a scatter diagram in Figure 3 shows the correlations between the CVA and the %MVC of Phase II in the neutral position. The %MVC of the extensors was significantly larger than that of the flexors in the neutral position (*p* < 0.001) (Figure 2). There was a negative correlation between the %MVC of the extensors in the neutral position (Phase II) and the CVA (*r* = −0.451, *p* < 0.001) (Figure 3), but not for the flexors (*r* = −0.164, *p* = 0.176).

### 3.2. Activities in the Cervical Extensors and Flexors in Each Motion Compared with the Neutral Position

The grand ensemble averages of the right and left for the extensors and flexors in flexion and extension showed the same pattern (Figure 1). For flexion, the %MVCs of the extensors in Phase I (*p* = 0.014) and Phase III (*p* < 0.001) and of the flexors in Phase II (*p* < 0.001) and Phase III (*p* < 0.001) significantly increased compared with the %MVC in the neutral position. For extension, the %MVCs of the extensors in Phase I (*p* = 0.020) and Phase II (*p* = 0.010) and of the flexors in Phase III (*p* < 0.001) significantly increased compared with the %MVC in the neutral position (Figure 2).

For each of the extensors and flexors, the right and left muscle activities showed the same pattern when viewed with respect to the direction of lateral flexion and rotation (Figure 1d–g). For the lateral flexion to the ipsilateral side, where the muscles are, the %MVCs of the extensors in Phase I (*p* = 0.006) and Phase II (*p* < 0.001) and of the flexors in every phase (*p* < 0.001 for each) significantly increased compared with the %MVC in the neutral position. For the lateral flexion to the contralateral side, where the muscles are, the %MVCs of the extensors (*p* = 0.018) and flexors (*p* = 0.004) in Phase III significantly increased compared with the %MVC in the neutral position (Figure 2).

For the rotation to the ipsilateral side, where the muscles are, the %MVCs of the extensors in every phase (*p* < 0.001 for each) significantly increased compared with the %MVC in the neutral position, but did not increase for the flexors in every phase. For the rotation to the contralateral side, where the muscles are, the %MVC of the extensors and flexors in every phase significantly increased compared with the %MVC in the neutral position (*p* < 0.001 for each) (Figure 2).

### 3.3. Comparison of the Cervical Extensors and Flexors Activities in Each Motion

For flexion, the %MVCs of the extensors were larger than those of the flexors in Phase I (*p* < 0.001) and Phase III (*p* < 0.001), but not for the flexors in Phase II (*p* = 0.060) (Figure 2). For extension, the %MVCs of the extensors in every phase were larger than those of the flexors (*p* < 0.001 for Phase I and II, *p* = 0.019 for Phase III) (Figure 2).

For lateral flexion, the %MVCs of the extensors in Phase I (*p* = 0.039) and Phase III (*p* < 0.001) in the ipsilateral side of the flexion, and the %MVCs of the extensors in every phase in the contralateral side, were significantly larger than the %MVCs of the flexors (*p* < 0.001 for each phase) (Figure 2).

For rotation, the %MVCs of the extensors in every phase in the ipsilateral side of rotation were larger than the %MVCs of the flexors (*p* < 0.001 for each phase), and the %MVCs of the extensors in Phase III in the contralateral side (*p* = 0.033) were significantly larger than the %MVCs of the flexors, but the %MVCs of the flexors in Phase I (*p* = 0.007) and Phase II (*p* < 0.001) in the contralateral side were significantly larger than those of the extensors (Figure 2).

### 3.4. Comparison in Activities of the Cervical Extensors and Flexors Activities among the Motions

For the extensors, the %MVCs were the largest in Phase I (*p* < 0.001) and Phase II (*p* < 0.001) of the rotation to the ipsilateral side, where the muscles are, and in Phase III (*p* < 0.001), the largest %MVC was in the flexion, compared with the %MVC of the other motions (Figure 2). For the flexors, the %MVCs were the largest in every phase of the rotation to the contralateral side, where the muscles are, compared with the %MVCs of the other motions (*p* < 0.001 for each phase) (Figure 2).

## 4. Discussion

In this study, sEMG was used to determine the activity of the cervical extensors and flexors in the neutral position and during flexion, extension, lateral flexion, and rotation of the neck in healthy participants. The %MVCs in the following were significantly greater than the %MVC in the neutral position: the extensors in flexion and extension, the ipsilateral extensors and the flexors in lateral flexion in Phase I; the flexors in flexion, and the extensors in extension and the ipsilateral extensors and flexors in lateral flexion in Phase II; the extensors and flexors in flexion, the flexors in extension, the bilateral flexors and the contralateral extensors in lateral flexion in Phase III; and the bilateral extensors and the contralateral flexors during rotation in all three phases.

### 4.1. Activities of the Cervical Muscles in Neutral Position

One of the main functions of the cervical muscles is to hold the head in posture [47]. The head and the cervical spine, which contributes to neck postural retention, are located in front of the center of gravity line in the neutral position [48]; therefore, the external torque in the direction of flexion due to gravity may be generated in the head and the cervical spine. The fact that the %MVCs of the cervical extensors were larger than that of the cervical flexors in the neutral position may be the result of the role of the cervical extensors, which are located on the posterior of the cervical spine, in maintaining the neutral position of the head and the cervical spine against the external torque in the direction of flexion due to gravity. To support our presumption that the cervical extensors play an important role in maintaining the neutral position of the head and neck, a negative correlation was found between the %MVC of the extensors and the CVA, but not for the flexors. It was reported that as the CVA decreases, the position of the second cervical vertebra with respect to the seventh cervical vertebra shifts forward and the horizontal distance between them increases [49], which means that the center of the head and the neck moves forward from the center of gravity line, which in turn leads to an increase in external torque with a decrease in the CVA. In other words, to maintain equilibrium against the increase in external torque in the direction of flexion due to the decrease in the CVA, the activity of the neck extensors increased, and a negative correlation between the %MVC and the CVA was observed. 

### 4.2. Cervical Muscle Activity in Each Motion Compared with the Neutral Position

In Phase I of flexion, the %MVCs of the flexors that could generate moments for the flexion direction [50] did not increase significantly from the %MVC of the neutral position. In contrast, the %MVC of the extensors increased significantly from the %MVC in the neutral position, and an eccentric contraction of the extensors was observed to support the head against gravity. These results suggest that flexion torque is produced mainly by gravity. In Phase II, the %MVCs of the flexors increased significantly from the %MVC of the neutral position to generate torque for flexion with gravity. The %MVCs of the extensors did not increase from the %MVC of the neutral position, despite the weight of the head acting as an external torque in the flexion direction. This finding supports the results of several previous studies that showed that the activity of the extensors decreases in the full flexion position [24,32,51]. In Phase I, the active postural holding was performed by the generation of extension moments of the extensors, which were significantly greater than those in the neutral posture; however, in Phase II, when the spine is held in full flexion, the posture is shifted to passive holding by structures such as the intervertebral discs, the ligaments, and the joint capsules [34,52]. In Phase III, returning the head to the neutral posture against gravity, the extensors activated concentric contraction with a significantly increased %MVC, compared to the %MVC in the neutral position. On the other hand, the flexors activated an eccentric contraction with a significant increase from the neutral position to adjust the speed of the motion of the head and neck to return from full flexion to the neutral position in three seconds.

For extension, the %MVC of the extensors that can generate moments in the direction of extension [50] increased significantly from the neutral position in Phases I and II; this finding clearly indicates that the activity of the extensors generates the extension torque. In contrast, there was no significant increase in the %MVCs of flexors from the %MVC in the neutral position, as observed in Phases I and II. This indicates that there was no gravitational extension torque on the head to induce muscle activity to hold the head position, because extension in Phase I begins in front of the center of gravity line; in Phase II of the full extension position, the weight of the head acts as an external torque in the direction of extension [53], and the torque to hold the head in this position is generated passively by spinal structures, such as the intervertebral discs, the ligaments, and the joint capsules, rather than active elements, which are activities of the flexors. In Phase III, concentric contraction of the flexors with an increase in %MVC occurred to return to the neutral position of the head from full extension against gravity. Thus, in the motion on the sagittal plane, torque was generated by the concentric contraction of the cervical muscles on the side of the neutral position during both flexion and extension from the full range of motion to the neutral position.

In Phases I and II of lateral flexion, both the %MVC of the extensors and flexors located ipsilateral to the direction of lateral flexion increased significantly from the neutral position, indicating that both the extensors and flexors jointly produced torque for the lateral flexion. This result supports the findings of a previous study that indicated that both the extensors and flexors ipsilateral to the direction of lateral flexion have moments of lateral flexion [50]. Ipsilateral rotation of all cervical vertebrae in a laboratory study [54] and ipsilateral rotation of cervical vertebrae except the axis in vivo study [55] were observed in cervical lateral flexion. However, in the results of this study, the %MVC of the contralaterally located flexors, which involved the generation of rotation torque in Phases I and II of rotation, showed no significant activity from the neutral position in Phases I and II of lateral flexion. These findings suggest that the cervical spine rotation that occurs during voluntary lateral flexion in the seated posture may not be due to muscle activity generating that torque. This presumption is supported by previous studies showing that rotation is generated by the oblique orientation of the zygapophysial joints of the cervical spine [55] and the presence of the Luschka joint [56], which suggests that the rotation associated with lateral flexion torque is passively generated. In Phase III, the %MVC of the extensors and flexors on the side of the direction to the neutral position from Phase II were significantly increased from the %MVC of neutral position, and both muscles were primarily responsible for generating lateral flexion torque to return from the full lateral flexion position to the neutral position. On the other hand, the flexors on the opposite side of the direction to the neutral position activated in eccentric contraction, with a significant increase from the neutral position, which may have arisen to adjust the speed of the motion of the head and neck to return from full lateral flexion to the neutral position within three seconds.

For rotation, the %MVC of the bilateral extensors and of the flexors located opposite to the direction of rotation were significantly increased from the neutral position in every phase. These results suggest that the bilateral extensors and the opposite flexors to the direction of rotation contribute to the generation of rotation torque. Comparing the activities of the extensors and flexors in Phase I—that is, the motion from the neutral position to the full range of motion, of flexion, extension, lateral flexion, and rotation—the activities of the extensors located on the side of direction of rotation and the flexors located on the opposite side of the direction of rotation were the largest. This indicates that both the extensors and flexors require the largest activity during rotation in the horizontal plane, compared to the other cervical motions in the other planes. There are three reasons for this result. First, motions on the sagittal and frontal planes benefit from external torque due to gravity, but motions on the horizontal plane cannot benefit from torque due to gravity. Therefore, on the horizontal plane, the motion must be carried out only with internal torque produced by cervical muscle contraction. Second, the muscle fibers of the cervical muscles are less likely to produce torque in motion on the horizontal plane than the other planes because they run almost vertically to the horizontal plane [57]. Finally, rotation has greater resistance from the spinal ligaments than other motions and, therefore, requires greater force to stretch the ligaments. This observation is supported by a previous report that severing the spinal articular capsule, pterygoid, and transverse ligaments increased the range of motion in rotation, but had less of an effect in flexion, extension, and lateral flexion [58].

### 4.3. Comparing the Activity of the Cervical Extensors and Flexors in Each Motion

Because the head and neck are located in front of the center of gravity line [48], the weight of the head acts as a flexion torque in the sitting position during motions in each direction from the neutral position. Even though each motion must be performed in such conditions, the involvement of the cervical ligaments in head stability is relatively small [14]; thus, the retention of the head position by cervical muscle activity is considered important [59]. The %MVCs of the extensors were greater than the %MVCs of the flexors in every phase of each motion, including the neutral position, except for Phases I and II of contralateral rotation. These results suggest that the role of holding the head is primarily played by the extensors rather than by the flexors. In other words, during cervical motions, not only is torque in each direction generated by the activities of the extensors and flexors but, in addition, the activities of the extensors are used to maintain stability while performing the motion. These characteristics seen in the activity of the cervical muscles may be related to the fact that the symptoms of NSNP are usually in the posterior neck rather than in the anterior neck [1].

### 4.4. Overall Interpretation of the Results

To obtain basic information for understanding the pathophysiology of neck pain and evaluating the effect of treatments of neck pain in the future, we measured the isotonic muscle activity of the cervical extensors and flexors in the neutral position and the neck motions on the three anatomical planes, using sEMG, in people without neck pain. The activity of the cervical muscles in the neutral position and the motions from the neutral position and the returning of the head to the neutral position from the maximum range of motion performed in this study are combinations of basic isotonic contractions of the cervical muscles under the influence of gravity-reflected motions in daily life. The results obtained in this study cannot be obtained from studies using sEMG observing isometric contractions. We believe that this point was a novelty in this study because, to our knowledge, there have been no reports of sEMG studies of motions with such a series of isotonic contractions that reflect the motions in daily life.

In previous studies, the activity of the trapezius, sternocleidomastoid, and splenius capitis muscles in isometric extension, flexion, and lateral flexion at the neck, using sEMG, was reported [4,5,22]. In the present study, isotonic muscle activity was obtained from the extensors and flexors as a muscle group, rather than from individual cervical muscles. This was done to simplify the measurement method so that sEMG could be applied in clinical practice with ease. On the other hand, in clinical practice it is possible to identify the cause of neck pain or abnormalities in cervical muscle activity caused by pain by multiple analyses, as shown in this study, such as simply looking at the balance between the extensors’ and the flexors’ activity at the neutral position, comparing the activities of the extensors and flexors between the neutral position and during each motion, or by looking at the balance between extensors’ and flexors’ activities during each motion. Thus, the sEMG-measuring performed in this study is relatively easy, and multiple analyses may identify the cause of pain or the abnormalities in muscle activity due to neck pain. However, in this study, the cervical muscles were not measured separately for each of the muscles, such as the trapezius and the sternocleidomastoid muscles observed in previous studies [4,5,22,28,29,30,31]. Instead, we observed the cervical muscles’ activities as those of a large group of the extensors and flexors, which made it impossible to examine the relationship between individual cervical muscle activity and the direction of motions, or the relationship between neck pain and individual cervical muscles.

Considering the above, we believe that the results of this study have the potential to be introduced into clinical practice in the future as a relatively simple method. The results of this study may also serve as a new objective index for evaluating neck pain in daily activities and the treatments of neck pain.

### 4.5. Limitations

Since the present study obtained the sEMGs of the cervical muscles in young healthy adults, the results of this study can be generalized for young adults. However, there is limitation in generalizing the results of this study for healthy adults of all ages, especially middle-aged to older adults. In addition, since the measurements were performed on Japanese people only, the results may not be generalizable to other ethnic groups, even in the younger age group. Future studies should include healthy middle-aged and older participants, since they comprise the most common age group of patients complaining of neck pain [60]. Another limitation is that only muscle activity in the sitting posture was recorded, which does not reflect cervical muscle activity in postures in which the effect of gravity is different, such as in the supine position; this limits the generalizability of the study results and is a research issue that must be resolved in the future, whereas it is essential to observe the muscle activity in the sitting position because we (especially office workers) spend much of our time in the sedentary position in our daily lives.

Despite these limitations, we believe that the cervical muscle activity patterns in the sagittal, frontal, and horizontal planes of motion in healthy participants identified in this study will provide reference data for understanding the pathophysiology of patients with neck pain and evaluating the effectiveness/efficacy of treatment for such patients. In the future, it is desirable to measure muscle activity in patients with NSNP, using the same methods used in this study, for comparison with the results of healthy people.

## 5. Conclusions

The activity of the bilateral cervical extensors and flexors in the neutral position and during flexion, extension, lateral flexion, and rotation in healthy adults, as shown in this study, may lead to a standardized pattern that can be used to understand the pathophysiology of neck pain and as an indicator for treatment evaluation.

In the neutral position, the %MVC of the extensors was significantly larger than that of the flexors.In the motion from the neutral position to the maximum range of motion (Phase I), the %MVCs of the extensors in flexion and extension, the ipsilateral extensors and flexors in lateral flexion were significantly larger than the %MVC in the neutral position.At the maximum range of motion (Phase II), the %MVCs of the flexors in flexion, the extensors in extension, and the ipsilateral extensors and flexors in lateral flexion were significantly larger than the %MVC in the neutral position.In the motion from the maximum range of motion to the neutral position (Phase III), the %MVCs of the extensors and flexors in flexion, the flexors in extension, the bilateral flexors and the contralateral extensors in lateral flexion were significantly larger than the %MVC in the neutral position.In rotation, the %MVCs of the bilateral extensors and the contralateral flexors in all three phases were significantly larger than the %MVC in the neutral position.

## Figures and Tables

**Figure 1 medicina-58-00728-f001:**
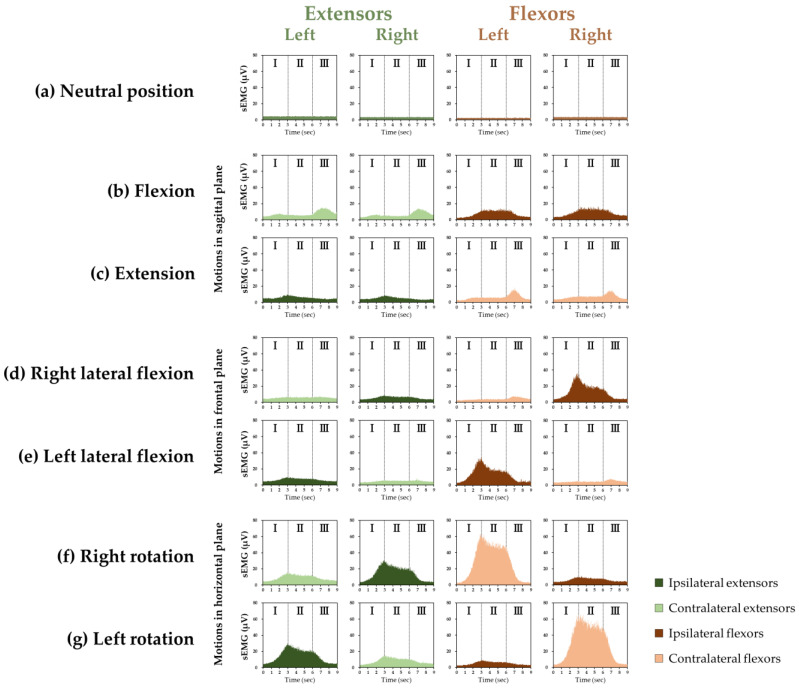
The grand ensemble average of surface electromyography (sEMG) of the cervical extensors and flexors in (**a**) the neutral position and during (**b**) flexion, (**c**) extension, (**d**) right lateral flexion, (**e**) left lateral flexion, (**f**) right rotation, and (**g**) left rotation. “I” indicates “Phase I”, a period (of three seconds) during a motion from the neutral position to the maximum range of motion. “II” indicates “Phase II”, a period (of three seconds) while maintaining the neck at the maximum range of motion. “III” indicates “Phase III”, a period (of three seconds) during a motion from the maximum range of motion to the neutral position. (**b**,**c**) show motion in the sagittal plane, (**d**,**e**) show motion in the frontal plane, and (**f**,**g**) show motion in the horizontal plane. The vertical and horizontal axes represent muscle activity and time, respectively. In (**a**) and in all the motions, the greenish color indicates the extensors and the brownish color indicates the flexors. In each motion, dark green indicates the ipsilateral extensors and dark brown indicates the ipsilateral flexors; light green indicates the contralateral extensors and light brown indicates the contralateral flexors.

**Figure 2 medicina-58-00728-f002:**
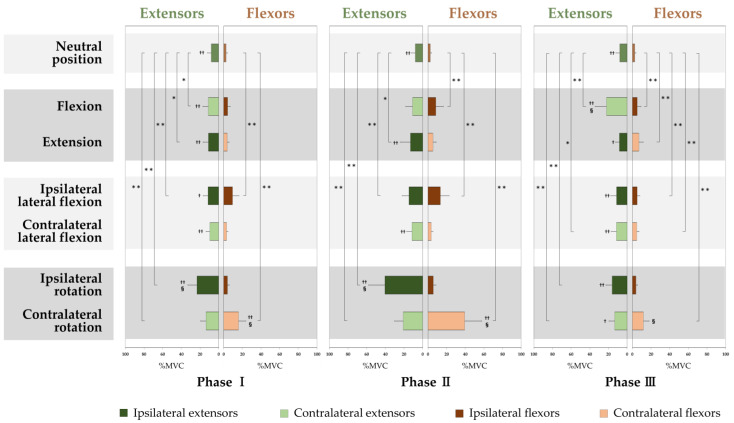
The percentage of maximal voluntary contractions (%MVC) (mean ± standard deviation) in each phase of the cervical extensors and flexors in each motion. There were significant differences in the %MVC when compared with the neutral position (* *p* < 0.05, ** *p* < 0.01) and between the corresponding extensors and flexors (^†^
*p* < 0.05, ^††^
*p* < 0.01). The largest %MVCs among all motions in each phase for the extensors and flexors, respectively, are indicated by “§” (*p* < 0.01).

**Figure 3 medicina-58-00728-f003:**
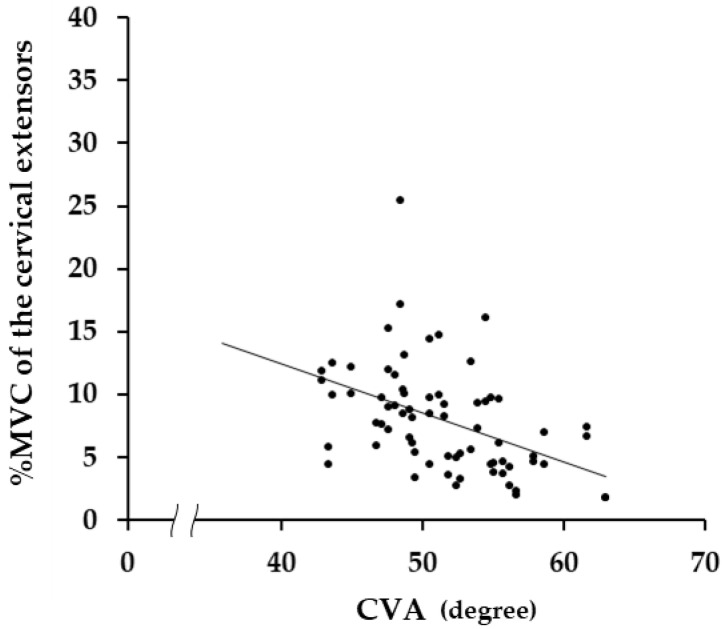
The correlations between craniovertebral angles (CVAs) and %MVCs in Phase II of the cervical extensors in the neutral position. There was a significant negative correlation between them (r = −0.451, *p* < 0.001).

## Data Availability

The data presented in this study are available on request from the corresponding author.

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
