# Peer review of "The Mode of Activity of Cervical Extensors and Flexors in Healthy Adults: A Cross-Sectional Study"

_medicina, 2022, doi:10.3390/medicina58060728_

Round 1
Reviewer 1 Report
Reviewer Comments
Thank you very much for the opportunity to review the manuscript submission entitled: "The mode of activity of cervical extensors and flexors in healthy adults."
The current paper aims to investigate the activity of bilateral cervical extensors and flexors during motion of the neck in the sitting position in healthy adults on the sagittal, frontal and horizontal planes, using a surface electromyogram (sEMG). The data is interesting, and it has a relevant rationale.
Specific comments
Title
- Include study design in the title.
Abstract
- Include the mean age of the participants in the abstract.
- Mentioning “asymptomatic” or healthy individuals is enough and not to mention without neck and shoulder symptoms.
- Include statistical results in the results of the abstract.
- Include MeSH terms as keywords
Introduction
- Need more emphasis on the Scientific background and explanation of the rationale why you need to do it the asymptomatic individuals and not in neck pain individuals.
- What is the clinical significance of this study? Use existing literature and you need to rewrite the introduction.
- Provide a hypothesis for your study
Methods
- Present key elements of study design early in the paper
- Describe the setting, locations, and relevant dates, including periods of recruitment and data collection
- Need to include more detailed inclusion and exclusion criteria and the sources and methods of selection of participants
- Describe any efforts to address potential sources of bias
- Explain how the study size was arrived at?
- What is the basis for your statistical tests? Your sample is low. Did the study data follow normal distribution? What statistical tests were used?
Discussion
- Give a cautious overall interpretation of results considering objectives, limitations, the multiplicity of analyses, and results from similar studies.
- Discuss the generalizability of the trial findings
Author Response
Thank you very much for your comments to improve our paper.
We send our resposes in the Word file.
Reviewer 2 Report
Paper has some criticisms that need a revision. Please look at these points:
- Lines 70-70: "we investigated sEMG with isotonic activities of the cervical extensors... to evaluate the effect of treatment on neck pain in this study." It is not clear how authors evaluate the effect of treatment. Drugs? Physiotherapy?
- Lines 36-38: " The pathoanatomical reason of NSNP is not well understood, although mechanical factors are more likely involved..." Improved this point. Which mechanisms? Microtrauma, congenital absence of cervical pedicle, cervical spondilosis. Look at these refs: -- doi: 10.4103/0028-3886.173669 -- doi: 10.1097/BRS.0000000000003873 -- doi: 10.1097/PRS.0000000000007484
- Figure 1 is good, but it is not well explained. Please improve the figure legend.
- Lines 310-312: "Ipsilateral rotation of all cervical vertebrae in vitro studies [49].. " Better is a laboratory study instead of "in vitro study"
- and ipsilateral rotation of cervical vertebrae except the axis in vivo studies [50] were observed in cervical lateral flexion".
- Lines 367-369: "Another limitation is that only muscle activity in the sitting posture was recorded, although it is essential to observe the muscle activity in the sitting position because we spend much of our time in the sedentary position in our daily lives, especially office workers" Please explain why authors did not record in supine position. Is there any reason?
- Lines 378-381. Conclusion does not reflect the whole paper. What this paper add new to the literature? Please report here their results.
Author Response
Thank you very much for your comments to improve our paper.
We send our responses in the Word file.
Round 2
Reviewer 1 Report
The Authors have addressed all of my concerns with the original manuscript. The revised manuscript is ready for publication
Reviewer 2 Report
Good.